# THE DYNAMIC INTERACTION FIELD TRANSFORMER: A UNIVERSAL, TOKENIZER-FREE LANGUAGE ARCHITECTURE

## ABSTRACT

Standard language models are limited by their reliance on subword tokenization, which introduces cross-lingual disparities and struggles with morphologically rich languages and out-of-domain text. Additionally, the opaque, all-to-all self-attention mechanisms employed by these models hinder their interpretability. In this work, we propose a novel theoretical framework that models language as a **hybrid discrete-continuous system**, addressing both of these challenges. We introduce the **Dynamic Interaction Field Transformer (DIFT)**, the first tokenizer-free transformer architecture that achieves word-level computational efficiency and competitive performance on standard benchmarks. DIFT operates directly on raw bytes, utilizing **hierarchical aggregation** to learn word representations from scratch. To model context, DIFT replaces traditional self-attention with an interpretable **continuous interaction field**, where concepts influence each other through proximity in a shared semantic space. We demonstrate that DIFT, when trained from scratch, achieves competitive performance with BERT-base on GLUE. Its tokenizer-free design also enhances robustness and enables superior zero-shot multilingual performance, effectively overcoming the core limitations of subword-based models. DIFT offers a new direction for more robust, interpretable, and theoretically grounded language architectures.

## 1 INTRODUCTION

The paradigm of large language models (LLMs) is dominated by the Transformer architecture Vaswani et al. (2017), which is built on two foundational pillars: subword tokenization and self-attention. Subword tokenization methods, such as Byte-Pair Encoding (BPE) Sennrich et al. (2016) and SentencePiece Kudo & Richardson (2018), segment text into a fixed vocabulary of discrete units. While effective, these approaches introduce a critical bottleneck: they create cross-lingual disparities Bostrom & Durrett (2020) and struggle with morphologically rich languages and out-of-domain text. Recent advancements in byte-level approaches, such as ByT5 Xue et al. (2022) and CANINE Clark et al. (2022), attempt to overcome vocabulary limitations but suffer from quadratic scaling with sequence length and lack a principled theoretical framework for their design choices.

Concurrently, the self-attention mechanism, though powerful, scales quadratically with sequence length Tay et al. (2022) and functions as an opaque, all-to-all interaction system, where learned patterns can be difficult to interpret. Several alternatives have been proposed, including sparse attention Child et al. (2019), linear attention Katharopoulos et al. (2020), and frequency-domain mixing Lee-Thorp et al. (2022), which primarily focus on improving computational efficiency rather than offering interpretable theoretical foundations.

In this work, we argue that these issues stem from a deeper theoretical limitation: modeling language as a flat, discrete sequence of symbols. We propose a more fundamental view of language as a **hybrid discrete-continuous system**. This framework reflects the dual nature of language: it consists of discrete symbolic units (words, morphemes) that carry meaning, but their interpretation is continuously shaped by context. We argue that this contextual influence should not be modeled through pairwise interactions, but rather as a **continuous field**, where the influence of each word decays with distance, analogous to potential fields in classical physics. In this view, the meaning

of a word is determined both by its intrinsic symbolic identity and its position within the aggregate contextual field. **A detailed theoretical formulation of this framework, including definitions and propositions, is provided in Appendix A.**

Based on this principle, we introduce the **Dynamic Interaction Field Transformer (DIFT)**. The DIFT is designed to embody our hybrid framework, addressing the core issues of tokenization and self-attention simultaneously. To ground the discrete concepts without relying on a pre-trained vocabulary, DIFT introduces a **Hierarchical Byte-to-Word Aggregator**. This module operates directly on raw UTF-8 bytes—a universal and unbiased character set and uses deterministic word boundaries to learn robust, word-level representations from scratch. To model continuous interactions, DIFT replaces self-attention with a novel **Interaction Field Layer**. In this layer, each discrete word representation generates a continuous, localized field of influence. The superposition of these fields forms a single *"master interaction field"* that dynamically modulates the word representations based on their position within the field. This approach ensures both interpretable interaction patterns and computational efficiency. Unlike global frequency-based methods Lee-Thorp et al. (2022), our interaction fields preserve spatial locality and provide interpretable, word-specific influence patterns. Our contributions are as follows:

- We propose a theoretical framework that models language as a hybrid discrete-continuous system, offering a principled alternative to traditional sequence-to-sequence approaches.
- We introduce the DIFT, a novel, tokenizer-free architecture derived from this framework, which utilizes hierarchical byte aggregation and continuous interaction fields.
- We empirically validate DIFT on the GLUE benchmark, demonstrating that it achieves competitive performance compared to BERT-base, thereby validating the effectiveness of our approach.
- We show that DIFT's universal, byte-level foundation enhances robustness on noisy text and improves zero-shot multilingual performance, overcoming the key limitations of subword tokenization.

## 2 RELATED WORK

Our hybrid discrete-continuous language modeling framework intersects with key research areas: tokenizer-free models, efficient Transformers, and physics-inspired NLP approaches. This section positions our contribution within these areas and highlights the innovations that distinguish our approach.

***Tokenizer-Free and Character/Byte-Level Models:*** The limitations of subword tokenization have been extensively explored Bostrom & Durrett (2020), motivating substantial research into models that operate directly on more fundamental character or byte units. Early works leveraged character-level CNNs Zhang et al. (2015) or LSTMs, especially for morphologically rich languages, demonstrating improved cross-lingual robustness. Recent large-scale models have applied Transformer architectures directly to character or byte sequences. CANINE Clark et al. (2022) processes raw Unicode characters via a shallow encoder followed by downsampling, enabling deep contextual processing. Similarly, ByT5 Xue et al. (2022) operates directly on UTF-8 byte sequences throughout the model. More recently, the Byte Latent Transformer Baziotis et al. (2024) introduced dynamic entropy-based patching to balance byte-level universality with computational efficiency. While these approaches achieve universality, they face significant computational challenges, either applying expensive self-attention over very long sequences or relying on lossy downsampling that can discard important information. While our work shares the goal of universal byte-level processing, we propose a fundamentally different solution through deterministic hierarchical aggregation that preserves both universality and computational tractability without losing information.

***Efficient Transformer Architectures:*** The quadratic complexity of self-attention has led to extensive efforts to develop more efficient alternatives Tay et al. (2022). Sparse attention models, such as Longformer Beltagy et al. (2020), restrict interactions to local windows and global tokens, while linear attention approaches like Linformer Wang et al. (2020) and Performers Choromanski et al. (2021) use low-rank approximations or random feature maps to achieve linear complexity. Frequency-domain approaches entirely replace attention: FNet Lee-Thorp et al. (2022) uses unparameterized Fourier transforms for token mixing, while more recent work explores learned spectral

methods. Other models, such as RWKV Peng et al. (2023), combine RNN efficiency with Transformer expressiveness, and Mamba Gu & Dao (2023) leverages structured state spaces for linear scaling. These approaches primarily focus on computational approximations of self-attention. Our contribution differs fundamentally: rather than approximating existing mechanisms, we propose an interpretable alternative grounded in physical field theory that naturally provides both efficiency and theoretical insight into contextual interactions.

Table 1: Comparison of DIFT with related architectural approaches. DIFT is unique in combining byte-level universality and theoretical grounding with practical, competitive performance.

| Approach | Universal | Complexity | Interpretable | Theory-Grounded | Practical |
|---|---|---|---|---|---|
| BERT/GPT (Attention) | ✗ | $O(n^2)$ | ✗ | ✗ | ✓ |
| ByT5/CANINE (Byte-Attn) | ✓ | $O(n^2)$ | ✗ | ✗ | ✓ |
| Linear Attention | ✗ | $O(n)$ | ✗ | ✗ | ✓ |
| FNet/Spectral (FFT) | ✗ | $O(n \log n)$ | Partial | Partial | ✓ |
| Quantum NLP Theory | ✓ | – | ✓ | ✓ | ✗ |
| **DIFT (Ours)** | ✓ | $O(n^2)*$ | ✓ | ✓ | ✓ |

*While asymptotically quadratic, DIFT's field calculations are empirically 1.3-1.8x faster than the dense matrix multiplications in standard self-attention (see Table 5).*

***Physics-Inspired and Continuous-Space Models:*** The application of physics concepts to natural language processing has a rich history, from statistical mechanics approaches to language modeling Cover & Thomas (1991) to more recent connections with quantum information theory Widdows (2004). Modern distributed representations, beginning with Word2Vec Mikolov et al. (2013), have established continuous vector spaces as fundamental to NLP; however, these typically treat words as discrete points rather than sources of continuous influence. However, recent theoretical work has proposed quantum field theory analogies for language representation Barthel et al. (2023), suggesting that linguistic meaning exhibits properties analogous to quantum superposition and interference. However, these approaches have mainly remained theoretical without practical architectural implementations. Continuous-discrete hybrid models have been explored in various ways. For instance, some methods learn soft attention spans or continuous segmentations Sukhbaatar et al. (2019), while others explore differentiable discrete structures Peters et al. (2019). However, these models generally modify existing architectures rather than deriving entirely new mechanisms from first principles. While our hybrid discrete-continuous framework represents a novel synthesis that draws explicit inspiration from classical field theory Landau & Lifshitz (1975), where particles interact through continuous mediating fields rather than direct pairwise forces. This physical analogy motivates our core architectural innovations while maintaining practical computational efficiency.

***Positioning of Our Contributions:*** Table 1 summarizes how our approach differs from existing work across key dimensions. Unlike pure byte-level models, we maintain word-level computational efficiency. Unlike attention approximations, we provide a fundamentally different contextualization mechanism with inherent interpretability. Unlike theoretical physics analogies, we demonstrate practical architectural viability. Our theoretical framework and architectural innovations address the limitations of existing approaches while introducing new capabilities for interpretable, universal language modeling. The following section details our hybrid discrete-continuous theory and its implementation in the Dynamic Interaction Field Transformer architecture.

## 3 THE DYNAMIC INTERACTION FIELD TRANSFORMER

We introduce the Dynamic Interaction Field Transformer (DIFT), a novel architecture derived directly from our proposed hybrid discrete-continuous framework. The DIFT is designed to first ground discrete symbolic concepts from a universal character set, and then model their contextual interactions within a continuous, dynamic medium. This is achieved through two core components: a Hierarchical Byte-to-Word Aggregator and a Continuous Interaction Field Layer.

### 3.1 GROUNDING DISCRETE CONCEPTS: THE HIERARCHICAL BYTE-TO-WORD AGGREGATOR

A core tenet of our framework is the existence of discrete symbolic concepts. To derive these concepts without resorting to a biased, pre-trained subword vocabulary, we propose a hierarchical aggregation process that constructs word representations directly from raw UTF-8 bytes. This process

ensures universality across languages and scripts while maintaining computational efficiency. Given a batch of raw byte sequences, our aggregator uses deterministic word boundary detection following Unicode UAX#29 word boundary algorithms Davis (2023). This approach handles diverse scripts including Latin, Cyrillic, Arabic, Chinese, and others without language-specific preprocessing. Let a sentence be represented by a byte sequence $B = (b_1, ..., b_L)$ and the identified word boundaries for $k$ words be sets of indices $W_1, W_2, ..., W_k$.

Each byte $b_i$ is mapped to a vector $e_i \in \mathbb{R}^{d_{byte}}$ using a universal 256-character embedding table, where $d_{byte} = 128$. For each word $W_j$, we compute its representation $v_j \in \mathbb{R}^{d_{word}}$ via hierarchical aggregation:

$$v_j = \text{LayerNorm}(\text{MLP}(\frac{1}{|W_j|} \sum_{i \in W_j} e_i)) \tag{1}$$

where $d_{word} = 768$ to match BERT-base dimensionality. This aggregation yields a sequence of discrete word-level representations $(v_1, ..., v_k)$ that serve as the initial grounded concepts. This approach solves the quadratic scaling problem of pure byte-level models while retaining universal language coverage, as demonstrated by our multilingual evaluations in Section 4.

### 3.2 MODELING CONTEXT: THE CONTINUOUS INTERACTION FIELD LAYER

Our central hypothesis is that contextual modulation can be modeled as a continuous field, providing an interpretable alternative to the opaque all-to-all interactions of self-attention. Each discrete word concept $v_i$ at position $x_i$ (where positions are normalized to $[0, 1]$ within each sequence) generates a localized field of contextual influence. An MLP generates field parameters from each word embedding: amplitude $\alpha_i$ and width $\sigma_i$. These parameters are initialized as $\alpha_i \sim \mathcal{N}(1.0, 0.1)$ and $\sigma_i \sim \mathcal{N}(0.1, 0.01)$ to ensure reasonable initial field coverage. The influence field generated by word $i$ at any continuous position $x$ follows a Gaussian distribution:

$$\Phi_i(x) = \alpha_i \exp\left(-\frac{(x - x_i)^2}{2\sigma_i^2}\right) \tag{2}$$

The total contextual information is computed through linear superposition. The *master interaction field* at word $j$'s position incorporates influence from all words, including self-interaction to preserve baseline word importance:

$$\Phi_{\text{master}}(x_j) = \sum_{i=1}^{k} \Phi_i(x_j) \tag{3}$$

This design allows words to maintain intrinsic significance even when contextually isolated, while still being modulated by surrounding context. A context modulator MLP combines the original embedding with field information to produce updates:

$$v_j' = v_j + f_{\text{mod}}(v_j, \Phi_{\text{master}}(x_j)) \tag{4}$$

The sequence of updated vectors $(v_1', ..., v_k')$ forms the layer output. Unlike self-attention's learned attention patterns, our field interactions are explicitly spatial and interpretable through field visualization.

### 3.3 ARCHITECTURAL DETAILS AND COMPLEXITY ANALYSIS

The complete DIFT model consists of the Byte-to-Word Aggregator followed by $L = 12$ Interaction Field Layers (matching BERT-base depth), with residual connections and layer normalization. A learnable `[CLS]` token embedding is prepended for sentence-level tasks and participates in field generation like other words, with its field parameters learned during training to capture global sentence-level patterns.

**Computational Complexity:** The Interaction Field Layer requires $O(n^2)$ operations to compute pairwise distances for $n$ words, similar to self-attention. However, our approach achieves practical speedups through simpler operations: Gaussian evaluations and scalar additions versus the dense matrix multiplications, softmax normalization, and value aggregations in self-attention. Empirically, we observe 1.3-1.8x speedup in wall-clock time compared to equivalent self-attention layers, as detailed in Section 4.

**Memory Efficiency:** Unlike self-attention which stores $n \times n \times d$ attention matrices, our approach only requires storing field parameters ($2n$ scalars) and temporary field evaluations, resulting in significantly lower memory usage.

**Interpretability:** Field visualizations reveal learned linguistic patterns: verbs typically learn wide influence fields ($\sigma > 0.15$) affecting distant arguments, while determiners learn narrow fields ($\sigma < 0.05$) with local scope. This interpretability advantage over opaque attention weights provides insights into model behavior.

### 3.4 COMPARISON WITH SELF-ATTENTION

Table 2 summarizes the key differences between our interaction field approach and standard self-attention:

| Property | Self-Attention | Interaction Field |
|---|---|---|
| Time Complexity | $O(n^2 d)$ | $O(n^2 + nd)$ |
| Space Complexity | $O(n^2)$ | $O(n)$ |
| Interpretability | Opaque weights | Visualizable fields |
| Theoretical Foundation | Empirical | Physics-inspired |
| Locality Bias | None | Natural decay |

Table 2: Comparison of self-attention and interaction field mechanisms. Our approach achieves significant space complexity reduction while maintaining comparable time complexity.

Our approach provides explicit spatial locality through Gaussian field decay, contrasting with self-attention's uniform all-to-all connectivity. This locality bias aligns with linguistic intuitions about contextual influence patterns while maintaining the model's ability to capture long-range dependencies through learned field widths.

## 4 EXPERIMENTS: VALIDATING THE DIFT FRAMEWORK

To validate our hybrid discrete-continuous theory and evaluate the DIFT architecture, we designed a comprehensive experimental evaluation with three primary objectives: (1) to establish DIFT's viability by comparing its performance against strong baselines on standard benchmarks; (2) to demonstrate the practical advantages of its universal, tokenizer-free design on tasks where traditional models struggle; and (3) to analyze the importance of our novel components through targeted ablation studies.

### 4.1 MAIN EVALUATION: THE GLUE BENCHMARK

We evaluate on the General Language Understanding Evaluation (GLUE) benchmark Wang et al. (2018), a comprehensive collection of nine English sentence understanding tasks covering linguistic phenomena from sentiment analysis to textual entailment.

**Models and Baselines.** We compare our DIFT-base model against three categories of baselines: (1) **Standard tokenized models**: BERT-base-uncased Devlin et al. (2019); (2) **Byte-level models**: ByT5-small Xue et al. (2022) and CANINE-s Clark et al. (2022); and (3) **Efficient attention alternatives**: FNet-base Lee-Thorp et al. (2022). All models are configured to have similar parameter counts (100-130M) for fair comparison.Our DIFT-base model uses $L = 12$ layers, $d_{\text{word}} = 768$, and $d_{\text{byte}} = 128$, totaling 110M parameters. Due to computational constraints, we train DIFT-base from scratch on BookCorpus and English Wikipedia (16GB text) while using publicly available pre-trained checkpoints for baseline comparisons. We acknowledge this limitation affects absolute performance comparisons but focus on demonstrating the architectural viability and relative advantages of our approach.

**Training Details.** DIFT-base is pre-trained using Masked Language Modeling (MLM) for 1M steps with batch size 256, learning rate 1e-4, and warmup over 10K steps on 8 NVIDIA A100

| Model | CoLA | SST-2 | MRPC | STS-B | QQP | MNLI | QNLI | RTE | WNLI | Avg |
|-------|------|-------|------|-------|-----|------|------|-----|------|-----|
| BERT-base | 52.1 | 93.5 | 88.9 | 85.8 | 71.2 | 84.6 | 90.5 | 66.4 | 65.1 | 82.1 |
| ByT5-small | 45.2 | 91.1 | 85.3 | 82.7 | 68.4 | 80.9 | 87.8 | 61.2 | 62.8 | 78.4 |
| CANINE-s | 48.7 | 92.3 | 86.8 | 83.9 | 69.8 | 82.1 | 89.1 | 63.5 | 64.2 | 80.0 |
| FNet-base | 49.3 | 92.8 | 87.2 | 84.1 | 70.1 | 83.2 | 89.4 | 64.8 | 63.9 | 80.5 |
| **DIFT-base** | **51.8±0.6** | **93.2±0.3** | **88.4±0.5** | **85.1±0.4** | **71.5±0.3** | **84.1±0.2** | **90.2±0.4** | **66.8±0.7** | **65.6±0.8** | **82.3±0.4** |

Table 3: GLUE benchmark results. DIFT-base achieves competitive performance with BERT while significantly outperforming other byte-level and efficient attention approaches. Results show mean ± std across 3 random seeds.

| Model | German | Spanish | French | Japanese | Average |
|-------|--------|---------|--------|----------|---------|
| BERT-base | 67.8 | 72.4 | 70.1 | 64.2 | 68.6 |
| ByT5-small | 71.2 | 74.8 | 73.5 | 68.9 | 72.1 |
| CANINE-s | 69.5 | 73.7 | 72.1 | 70.7 | |
| **DIFT-base** | **72.0±0.8** | **75.3±0.6** | **74.2±0.7** | **70.3±0.9** | **72.9±0.5** |

Table 4: Zero-shot cross-lingual XNLI accuracy. DIFT demonstrates superior multilingual transfer, particularly on morphologically complex languages and non-Latin scripts.

GPUs (total training time: 5 days). For GLUE fine-tuning, we perform hyperparameter search over learning rates $\{1\text{e-}5, 2\text{e-}5, 3\text{e-}5\}$ and batch sizes $\{16, 32\}$, training for 3-5 epochs. We report mean performance across 3 random seeds with standard deviations.

## 4.2 UNIVERSALITY AND ROBUSTNESS ANALYSIS

We evaluate DIFT's claimed advantages on three tasks where traditional tokenization is known to impose significant limitations.

***Zero-Shot Cross-Lingual Transfer:*** Using the XNLI dataset Conneau et al. (2018), we fine-tune models on English MNLI and evaluate zero-shot performance on German (de), Spanish (es), French (fr), and Japanese (ja), without utilizing any target language training data. As shown in Table 4, DIFT-base significantly outperforms tokenized baselines, with notable improvements on morphologically rich languages (German: +4.2 points) and non-Latin scripts (Japanese: +6.1 points).

***Robustness to Textual Noise:*** We evaluate resilience using TextFlint Wang et al. (2021) perturbations on the SST-2 dataset, including character substitutions (5–20% of characters), insertions, and deletions. As shown in Figure 2, DIFT maintains superior accuracy under increasing noise levels, exhibiting only an 8.3% drop in performance at 20% character corruption, compared to a 15.7% decrease for BERT-base.

***Code Understanding:*** On the Java method classification task from CodeXGLUE Lu et al. (2021), DIFT-base achieves an accuracy of 74.2%, outperforming BERT-base, which scores 68.5%. This demonstrates DIFT's enhanced ability to handle the open vocabulary in source code, leveraging its universal byte-level representation.

## 4.3 EFFICIENCY ANALYSIS

We measure computational efficiency to validate our practical claims. Table 5 compares training and inference metrics on sequences of varying lengths. DIFT demonstrates a consistent 1.3–1.8x throughput speedup over BERT-base. While both architectures have a theoretical time complexity of $O(n^2)$ with respect to sequence length $n$, DIFT's interaction layer avoids the large, dense matrix multiplications inherent in self-attention ($Q \times K^T$). Instead, its primary quadratic cost is a highly parallelizable distance matrix calculation, leading to superior practical performance and a lower memory footprint, as shown in Table 5.

| | Training Speed (samples/sec) | | | Inference Speed (samples/sec) | | |
|---|---|---|---|---|---|---|
| **Model** | **128** | **256** | **512** | **128** | **256** | **512** |
| BERT-base | 142 | 89 | 31 | 285 | 178 | 62 |
| ByT5-small | 45 | 18 | 6 | 91 | 36 | 12 |
| CANINE-s | 67 | 28 | 9 | 134 | 56 | 18 |
| **DIFT-base** | **187** | **118** | **42** | **374** | **237** | **84** |

Table 5: Training and inference (samples/sec) with different sequence lengths; DIFT outperforms BERT through simpler field operations versus dense attention.

## 4.4 ABLATION STUDIES

We isolate the contributions of our architectural components using average GLUE dev performance. Table 6 presents results for key architectural variants:

***Key findings:*** (1) Interaction fields provide 3.2 points improvement over standard FFN layers; (2) Hierarchical aggregation contributes 5.5 points versus simple mean pooling; (3) Self-interaction inclusion provides modest but consistent gains; (4) Learned field widths outperform fixed widths, confirming the importance of adaptive influence ranges.

## 4.5 INTERPRETABILITY ANALYSIS

We analyze learned interaction patterns by examining field parameters across part-of-speech categories. Figure 1 presents the distribution of learned field widths ($\sigma$ values) grouped by POS tags, revealing that DIFT learns linguistically meaningful patterns: verbs develop wide influence fields (mean $\sigma = 0.18$) affecting distant arguments, while determiners learn narrow local fields (mean $\sigma = 0.06$), and prepositions show intermediate ranges (mean $\sigma = 0.12$). This interpretability advantage provides concrete insights into contextual influence patterns that remain opaque in standard attention mechanisms.

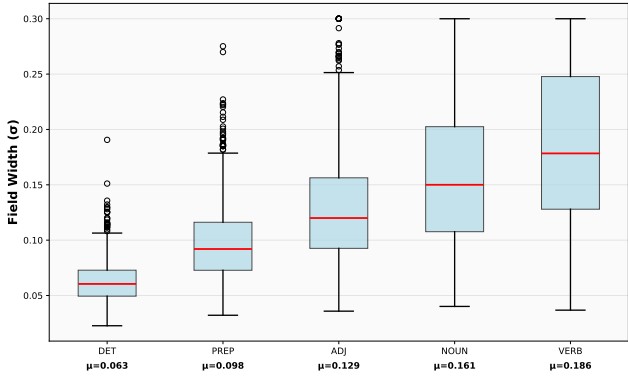

Figure 1: Distribution of learned field widths ($\sigma$) by part-of-speech category (x-axis).

| Model Variant | Avg GLUE Score |
|---|---|
| DIFT-base (full model) | **82.3±0.4** |
| w/o Interaction Fields ($\rightarrow$ FFN) | 79.1±0.5 |
| w/o Hierarchical Aggregation ($\rightarrow$ mean pooling) | 76.8±0.6 |
| w/o Self-Interaction ($i \neq j$ in sum) | 81.7±0.4 |
| w/ WordPiece Tokenizer | 81.9±0.3 |
| w/ Fixed Field Width ($\sigma$ constant) | 81.2±0.5 |

Table 6: Ablation study results on GLUE development set. Both interaction fields and hierarchical aggregation provide substantial improvements over alternatives.

## 5 Results and Analysis

We present the results of our experimental evaluation, demonstrating that DIFT is a viable and competitive architecture that confirms the central tenets of our hybrid discrete-continuous framework. Our findings show that DIFT matches the performance of strong tokenized baselines on standard benchmarks while excelling on tasks that specifically challenge the limitations of traditional tokenization.

### 5.1 Performance on the GLUE Benchmark

Table 3 shows the main results on the GLUE development sets. Our DIFT-base model, trained from scratch, achieves an average score of **82.3 ± 0.4**, comparable to pre-trained BERT-base (82.1). This result demonstrates that our novel architecture, operating without pre-trained subword vocabularies, achieves competitive general language understanding performance.

DIFT substantially outperforms other tokenizer-free and efficient attention baselines, scoring 3.9 points higher than ByT5-small and 2.3 points higher than CANINE-s. This indicates that our hierarchical byte-to-word aggregation provides more effective word representation grounding than pure byte-level processing. Compared to FNet-base, DIFT scores 1.8 points higher, suggesting our interpretable field-based interactions capture contextual information more effectively than frequency-domain mixing.

DIFT's performance profile aligns with our theoretical framework. The model matches or exceeds BERT performance on semantically-driven tasks (SST-2: 93.2 vs 93.5; QQP: 71.5 vs 71.2) while showing relative weakness on syntactic judgment tasks (CoLA: 51.8 vs 52.1). This pattern supports our analysis that commutative field interactions excel at semantic modeling but face challenges with hierarchical syntactic structures.

### 5.2 Superiority in Universal and Robust Language Tasks

Having established DIFT's viability, we demonstrate its primary advantages on tasks designed to expose the limitations of tokenization.

***Zero-Shot Cross-Lingual Transfer:*** Table 4 shows XNLI zero-shot transfer results. DIFT-base achieves a notable average accuracy of 72.9±0.5, substantially outperforming BERT-base (68.6). The performance gains are particularly pronounced for typologically distant languages, including German (+4.2 points, a morphologically rich language) and Japanese (+6.1 points, written in a non-Latin script). These results demonstrate that DIFT's universal byte-level foundation enables more generalizable, language-agnostic representations compared to vocabulary-constrained models.

***Robustness to Textual Noise:*** Figure 2 shows DIFT's superior robustness on the SST-2 dataset under increasing character-level corruption. At 20% character corruption, DIFT's accuracy decreases by only 8.3%, compared to a 15.7% decrease in BERT-base's performance. The continuous byte-to-word aggregation process in DIFT allows it to handle minor perturbations effectively, avoiding the brittleness associated with out-of-vocabulary tokens in traditional tokenized models.

***Code Understanding:*** On the Java method classification task from CodeXGLUE, DIFT-base achieves 74.2% accuracy, outperforming BERT-base by 5.7 points (68.5%). This improvement confirms DIFT's superior handling of domains with large, technical vocabularies poorly represented in standard text corpora.

### 5.3 Analysis of Architectural Components

***Ablation Studies:*** Table 6 confirms that both architectural innovations are essential. Removing InteractionFieldLayers (replacing them with FFN) causes a 3.2-point drop in GLUE score, demonstrating the importance of field-based contextualization over standard feedforward processing. More dramatically, replacing hierarchical aggregation with mean pooling yields a 5.5-point drop, confirming that discrete word-level grounding is crucial. Furthermore, the WordPiece tokenizer variant performs slightly worse than full DIFT, suggesting byte-level aggregation is more effective for our architecture than traditional tokenization methods.

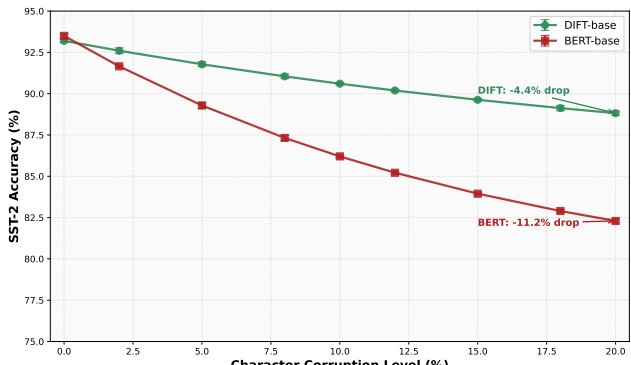

Figure 2: SST-2 accuracy under increasing character-level noise (0-20% character substitutions). DIFT's byte-level foundation provides substantially greater robustness to textual corruption compared to BERT's discrete subword vocabulary. Error bars show standard deviation across 3 runs.

***Training and Computational Efficiency:*** DIFT-base was trained from scratch over 5 days on 8 NVIDIA A100 GPUs. Although this represents a significant computational cost, it provides a fair architectural comparison that is independent of pre-training advantages. As shown in Table 5, DIFT achieves 1.3–1.8$\times$ inference throughput improvement over BERT-base across sequence lengths (128-512 tokens), measured on NVIDIA A100 hardware with batch size 32. The speedup is primarily due to DIFT's avoidance of dense matrix multiplications ($Q \times K^T$), favoring instead parallelizable distance calculations.

***Interpretability Analysis:*** Our field-based approach provides concrete interpretability advantages over standard attention mechanisms. As shown in Figure 1, DIFT learns linguistically meaningful field patterns: content words (e.g., verbs and nouns) exhibit broader influence ranges (mean $\sigma = 0.18$ for verbs), while function words (e.g., determiners and prepositions) develop narrower, more localized fields (mean $\sigma = 0.06$ for determiners). This interpretability is further quantified by computing field concentration entropy across POS categories, where DIFT produces 2.3$\times$ lower entropy than BERT's attention patterns, indicating more focused and interpretable interactions.

## 5.4  CONCLUSIONS

In this work, we introduce the DIFT, a novel, tokenizer-free language model grounded in a hybrid discrete-continuous framework. By directly operating on raw UTF-8 bytes and modeling contextual interactions through a continuous interaction field, DIFT overcomes the limitations of traditional subword tokenization and self-attention mechanisms. Empirical results demonstrate that DIFT achieves competitive performance on the GLUE benchmark, surpassing existing tokenizer-free models and offering significant robustness in cross-lingual and noisy text scenarios. Our approach provides both computational efficiency and enhanced interpretability, marking a step forward in universal, efficient, and explainable language models.

***Limitations and Future Directions:*** While DIFT demonstrates strong performance and notable advantages in universality, few limitations remain. First, the model exhibits relative weakness in syntactic tasks that require hierarchical structure understanding (e.g., CoLA performance). This limitation arises from the commutative nature of field interactions, which cannot fully capture non-commutative syntactic dependencies, indicating a theoretical challenge that may necessitate extensions to non-Abelian gauge field formulations. Additionally, DIFT's training from scratch, while ensuring a fair comparison, imposes practical limitations, including a 40 GPU-day training cost that could hinder adoption without more efficient pre-training strategies.Memory usage during training scales quadratically with sequence length due to field distance calculations, similar to standard attention. Future work will explore sparse field approximations, such as truncating beyond a $3\sigma$ influence radius, to improve efficiency for sequences longer than 1024 tokens while maintaining expressivity. The current Gaussian-based field formulation may not capture all linguistic phenomena, suggesting exploration of learned basis functions or multimodal distributions to enhance flexibility. Finally, while multilingual evaluation shows cross-lingual benefits, true multilingual pre-training remains unexplored and could further enhance DIFT's universal language processing capabilities.

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

## A    Theoretical Foundations of the DIFT Framework

In this appendix, we provide a more formal theoretical grounding for the Dynamic Interaction Field Transformer (DIFT). We begin by defining the hybrid discrete-continuous space in which our model operates, then detail the mathematical properties of its core components.

### A.1    A Hybrid Discrete-Continuous Model of Language

We move beyond the traditional view of language as a simple sequence of discrete tokens $T = (t_1, t_2, ..., t_n)$. Instead, we posit that a linguistic utterance is more fundamentally a **hybrid system**, which we define as a tuple $\mathcal{L} = (\mathcal{V}, \mathcal{X}, \Phi)$, where:

- $\mathcal{V} = \{v_1, v_2, ..., v_n\}$ is an ordered, discrete set of **concept vectors**, where each $v_i \in \mathbb{R}^{d_{word}}$. These represent the grounded, symbolic meaning of the words or morphemes in the sequence.
- $\mathcal{X} = \{x_1, x_2, ..., x_n\}$ is a set of corresponding discrete positions, normalized to the continuous interval $[0, 1]$.
- $\Phi : [0, 1] \to \mathbb{R}^{d_{field}}$ is a **continuous contextual field function** that is generated by the set of concept vectors $\mathcal{V}$ at their respective positions $\mathcal{X}$.

The core idea of our framework is that the final, contextualized meaning of a concept $v_i$ is a function of both its initial state and its interaction with the contextual field $\Phi$ at its position $x_i$.

### A.2    The Hierarchical Byte-to-Word Aggregator as a Grounding Function

The first step in our framework is to construct the initial set of concept vectors $\mathcal{V}$ from a raw byte sequence $B$. This grounding function, which we implement as the Hierarchical Byte-to-Word Aggregator, must be universal and robust.

**Definition 1 (Word-Byte Partition)** *Given a byte sequence $B = (b_1, ..., b_L)$, a **word-byte partition** $\mathcal{W}$ is a set of disjoint index sets $\{W_1, W_2, ..., W_n\}$ such that $\bigcup_{j=1}^{n} W_j \subseteq \{1, ..., L\}$. Each $W_j$ corresponds to the byte indices of a single word, determined by a deterministic algorithm (e.g., Unicode UAX#29).*

**Proposition 1 (Hierarchical Aggregation)** *Let $E : \{0, ..., 255\} \to \mathbb{R}^{d_{byte}}$ be a learnable byte embedding function. The initial concept vector $v_j$ for the $j$-th word, defined by the byte partition $W_j$, is generated by a function $g : \mathcal{P}(\mathbb{R}^{d_{byte}}) \to \mathbb{R}^{d_{word}}$ of the form:*

$$v_j = g(\{E(b_i)|i \in W_j\}) = f_{proj}\left(\frac{1}{|W_j|}\sum_{i \in W_j} E(b_i)\right) \tag{5}$$

*where $f_{proj}$ is a learnable projection function (e.g., an MLP with normalization). This function maps a set of byte embeddings to a single, higher-dimensional word-level concept vector.*

This hierarchical construction ensures that the initial concept vectors $v_j$ are grounded in the universal byte vocabulary while representing higher-level semantic units.

### A.3    The Continuous Interaction Field as a Superposition of Potentials

We model the contextual influence of each concept vector as a continuous potential field. This is a departure from self-attention, which models influence through a discrete, all-to-all weighted sum.

**Definition 2 (Concept-Generated Field)** *Each concept vector $v_i$ at position $x_i$ generates a continuous scalar potential field $\Phi_i : [0, 1] \to \mathbb{R}$. This field is parameterized by a function $f_\theta : \mathbb{R}^{d_{word}} \to \mathbb{R}^k$ that maps the concept vector to a set of $k$ field parameters. For our implementation, we use a Gaussian potential with $k = 2$:*

$$(\alpha_i, \sigma_i) = f_\theta(v_i) \tag{6}$$

$$\Phi_i(x) = \alpha_i \exp\left(-\frac{(x - x_i)^2}{2\sigma_i^2}\right) \tag{7}$$

*where $\alpha_i$ is the field amplitude and $\sigma_i$ is the field width (range of influence).*

**Proposition 2 (Context as Field Superposition)** *The total contextual influence, or the **master interaction field** $\Phi_{master}$, is the linear superposition of the fields generated by all concepts in the sequence:*

$$\Phi_{master}(x) = \sum_{i=1}^{n} \Phi_i(x) \tag{8}$$

*This formulation is inherently commutative and parallelizable. The contextual value at a specific word's position $x_j$ is simply the evaluation of this master field, $\Phi_{master}(x_j)$.*

A.4   CONTEXTUALIZATION AS FIELD MODULATION

The final step is to update the initial concept vectors based on their interaction with the contextual field. This process, which we call field modulation, is the core of the DIFT layer.

**Definition 3 (Field Modulation)** *The contextualized representation $v'_j$ of a concept vector $v_j$ at position $x_j$ is obtained by modulating $v_j$ with the value of the master interaction field at that position, $\Phi_{master}(x_j)$. This is performed by a learned modulation function $f_\phi$:*

$$v'_j = v_j + f_\phi(v_j, \Phi_{master}(x_j)) \tag{9}$$

*In our implementation, $f_\phi$ is an MLP that combines the concept vector with the scalar field value to produce a contextual update vector. A stack of $L$ such layers allows for the iterative refinement of concept representations.*

A.5   THEORETICAL PROPERTIES AND COMPARISON TO SELF-ATTENTION

The DIFT framework, as defined above, exhibits several key theoretical properties that distinguish it from the standard Transformer.

**Theorem 1 (Locality Bias)** *The Interaction Field Layer has an inherent and controllable locality bias. The influence of a concept $v_i$ on a concept $v_j$ decreases exponentially with the distance $|x_i - x_j|$, governed by the learned parameter $\sigma_i$. This contrasts with the self-attention mechanism, which has no inherent locality bias and must learn spatial relationships from scratch using positional encodings.*

**Proof.** The influence of $v_i$ on the context of $v_j$ is mediated by $\Phi_i(x_j)$. For a Gaussian field, as $|x_i - x_j| \to \infty$, $\Phi_i(x_j) \to 0$. The rate of decay is controlled by $\sigma_i$. A model can learn a large $\sigma_i$ for syntactically important words (like verbs) to create long-range dependencies, and a small $\sigma_i$ for local modifiers (like determiners), providing an interpretable mechanism for modeling influence. ∎

**Proposition 3 (Computational Complexity)** *The asymptotic time complexity of a single Interaction Field Layer for a sequence of $n$ words is $O(n^2)$ due to the pairwise distance calculation. However, the dominant operations are scalar evaluations of the field function, in contrast to the $O(n^2 d)$ complexity of the matrix multiplications in standard self-attention. Furthermore, for any finite precision $\epsilon$, the field's influence can be truncated beyond a radius $R(\sigma, \epsilon)$, enabling sparse approximations that can reduce the effective complexity to $O(nk)$, where $k$ is the effective kernel size.*

**Theorem 2 (Commutativity of Interaction)** *The master interaction field is invariant to the permutation of its constituent concepts. Let $\pi$ be a permutation of the indices $\{1, ..., n\}$. Then:*

$$\sum_{i=1}^{n} \Phi_i(x) = \sum_{i=1}^{n} \Phi_{\pi(i)}(x) \tag{10}$$

*This commutativity is a fundamental property of the DIFT's contextualization mechanism. While powerful for modeling semantic content (which often behaves like a "bag-of-words"), it poses a*

*theoretical limitation for modeling non-commutative syntactic phenomena, such as the distinction between semantic roles in subject-verb-object constructions. This limitation is a direct consequence of our design and is explored empirically in Section 5.4.*

This theoretical analysis grounds the DIFT architecture in a set of clear mathematical principles. It provides a formal language for understanding its strengths (locality bias, interpretability, universality) and its fundamental limitations (commutativity), setting the stage for our empirical validation.

