# OpenReview forum: "The Dynamic Interaction Field Transformer: A Universal, Tokenizer-Free Language Architecture"
_ICLR.cc/2026/Conference — ICLR 2026 Conference Withdrawn Submission_

### Official Review · Reviewer_jQq5 · 2025-10-23

**Soundness:** 2
**Presentation:** 2
**Contribution:** 3
**Rating:** 4
**Confidence:** 4

**Summary:**

The manuscript introduces a novel, physics-inspired framework for language modeling by that addresses two key limitations of current architectures, namely the reliance on fixed tokenization schemes and the opacity of the all-to-all attention mechanism.
The proposed model replaces the former with a hierarchical byte-to-word aggregation process grounded in Unicode-based character embeddings, rendering it inherently language-independent. The latter is reformulated through a field-theory or kernel-based approach which explicitly accounts for locality. Overall, the algorithm achieves performances comparable to BERT-base while offering enhanced interpretability of words interactions. Furthermore, the framework demonstrates promising cross-lingual generalization, despite being trained exclusively on English text.

**Strengths:**

1. the byte-to-word encoding, to me, represents the main and most interesting result of the paper. Given the ablation experiments, it also seems to represent the main ingredients in terms of performance.
2. the idea of using physics knowledge for modeling words interactions is interesting and  the framework looks promising
3. the paper represents a proof of concepts that could be scaled

**Weaknesses:**

1. The presentation, despite being appealing and engaging, needs to be improved:
	a. paragraph 5.2 is identical to 4.2; paragraph 5.3 contains the same information presented in paragraphs 4.3, 4.4 and 4.5
	b. There are a lot of missing details concerning the implementations of the algorithm, namely: what are the specific (number of layers, hidden dimensions, activation functions) of $f_{proj}$in eq 5, $f_\theta$ in eq 6, $f_\phi$ in eq 9? How do you invert eq 1 to associate a word $W_i$ to a representation $v_i$ after it was processed (i.e. the equivalent of the unembedding matrices)
	c. theorem 1 in A.5 cannot be considered a theorem
2. The language and computational performances improvements over BERT are marginal, meaning that I would expect the authors to dwell more on the repeatedly mentioned, but only very briefly discussed in sec 4.5, topic of interpretability, for instance by discussing also the meaning of $\alpha_i$ coefficients and/or investigating properties of the representations $v_i$.
3. As mentioned by the authors, the paper would largely benefit from a inspecting, even if just partially, different functional form of the fields. Would an exponential distribution work equally likely? What if they use fat-tail/power law distributions like the Cauchy one?

**Questions:**

1. concerning the discrete-continuous discussion about language, I'd find useful to mention: Boleda, G. 2025. [LLMs as a synthesis between symbolic and continuous approaches to language](https://arxiv.org/pdf/2502.11856).
2. Given this sentence: "Mark went to the shopping mall on Wednesday evening with Jade and Caroline. When did you see him last?" Would "Mark" interact with "him" with you locality bias? A $\sigma$ of 0.2 would not allow the two fields to touch and influence each other when they are so far away in the sentence.
3. Is the model really capable of providing high performances on code while being trained on bookcorpus and wikitext?

Depending on how weaknesses and questions are addressed, I would be willing to change my score

---

### Official Review · Reviewer_3TFN · 2025-10-29

**Soundness:** 2
**Presentation:** 2
**Contribution:** 3
**Rating:** 2
**Confidence:** 4

**Summary:**

This work proposes a new architecture composed of two novel primitives, namely a tokenization-free embedding method and a sequence-mixer, as an alternative to Transformer and softmax attention. Effectively, this sequence mixer models interactions between tokens based primarily on their relative distance in the sequence, as opposed to modeling token embeddings’ similarity in softmax attention. Surprisingly, this method works on par or better than Transformer in small-scale (around 110M parameters) bidirectional language model validation on GLUE benchmark.

The ideas in the paper are interesting but they require more rigorous and detailed research, presentation and validation (see weaknesses). If the problems, especially with insufficient presentation and unfounded claims are resolved, I'm willing to reevaluate the work.

**Strengths:**

* Both architectural elements introduced are novel and interesting concepts. From my understanding, the new sequence-mixing mechanism deviates far enough from standard attention and is more light-weight.

* The new architecture is significantly faster than parameters-matched Transformer despite having the same asymptotic run-time complexity.

* The interactions between tokens in the sequence-mixing mechanism of the DIFT are dependent on the distance between their positions and a representation of **just one of the tokens**. In the baseline BERT Transformer, interactions depend on the similarity of high-dimensional representations of **both** tokens, which in turn already have positional information incorporated. From my point of view, the Transformer interaction modeling paradigm is more powerful. Yet it is even more remarkable and even surprising that this model shows similar modeling performance to parameters-matched BERT on the established GLUE benchmark.

* The proposed tokenizer-free method of creating token embeddings seems to benefit performance on several tasks, such as XNLI.

**Weaknesses:**

**Insufficient descriptions**

DIFT algorithm is not fully specified. For example, neither lines 196-199, nor 668-670 establish a precise mathematical formula for $f_{\phi}$. Likewise, there are no formulas for $\alpha_i$ and $\sigma_i$ params. It doesn’t help that there’s no model code listing in the Appendix or accompanying code archive in the submission.

The paper positions DIFT against Transformer-based models, conceptually and empirically. But there is no objective comparison of the method with self-attention Transformer architecture that would highlight their similarities and differences. Ideally, the two architectures could be described side-by-side by their respective sets of equations, using compatible notation.


The exposition of the novel embeddings formation method and its distinctions from other methods is also rudimentary. In particular, lines 160-167 don’t offer a rigorous description of how word boundaries are formed from byte sequences and what are the differences from established methods like BPE or WordPiece. Moreover, the distinctions and advantages of the method in relation to other tokenization-free models, such as recent ByteLatentTransformer [6] and H-Net [7], are not clearly stated.

Equation 1 doesn’t explain why the embedding method is called “hierarchical aggregation” – it just calculates mean over representations of individual bytes in a word.
Experimentation.

Many hyper-parameters are not specified. Specifically, sequence length, learning rate schedule, optimizer and its parameters, numerical precision, software frameworks used (e.g., PyTorch or JAX, HuggingFace, Lightning, DeepSpeed, etc.) and optimizations (custom CUDA or Triton kernels, CUDA Graphs, compilation, etc.). The lack of hyper-params reporting, especially sequence length or total number of tokens for pre-training, might cast doubts about the fairness of comparisons in Table 3, because the other models were pre-trained on about 100B tokens each.

**Experimentation**


DIFT combines two distinct architectural innovations – initial embedding creation and sequence-mixing mechanism. However, it is not clear to what extent each of them contributes individually to performance and efficiency of the model. Ablations where only embeddings and only self-attention are replaced in Transformer model by novel modules would help determine their individual effectiveness.

The scope of empirical validation is limited and could be expanded. Currently, the main result is pre-training of an encoder BERT-like model and its fine-tuning on GLUE. However, GLUE benchmark tests performance on short sequences only ($\leq 256$ with majority of samples having length $\leq 128$, [8]). Since DIFT has natural locality bias (line 235), it remains unclear whether the performance translates to longer sequences. At least results on perplexity on longer contexts could be provided and compared to other models.

I'd recommend to validate the model on some of these additional modalities: autoregressive language modeling, image classification, speech recognition, long-range dependencies modeling and synthetic benchmarks. The appeal of byte-level tokenizer-free models lies in their ability to process heterogenous data (e.g., images and text simultaneously), but now all but text remains unexplored.

Good examples of comprehensive experimentations with a limited compute budget are presented in papers [9-10], and FNet which is cited in the paper.

**The claims about self-attention’s lack of theoretical foundation and interpretability are unfounded.**

The recurring claim throughout the paper is that different self-attention mechanisms are “opaque”, uninterpretable, and not theory-grounded. I beg to differ in the matters of standard attention’s interpretability and groundedness.

There is a vast literature on the interpretability of self-attention. I would recommend to review that literature before making such claims. A good starting point is papers [1-2], and you can proceed with subsequent papers which cite [1-2].

I’d like to add that attention, either softmax or linear one, has a very simple interpretation – for any token (or “word” in the terminology of this work), attention scores show how much each other token in the sequence should influence the next-layer representation of that token. In other words, attention scores show *importances* of tokens, given the similarity of their representations in a high-dimensional space (dot product of **q** and **k**).

Regarding theoretical foundation, one could argue that attention draws from psychology or the science of how the human brain works – see [4, 5] for example.

“Unlike self-attention’s learned attention patterns, our field interactions are explicitly spatial and interpretable through field visualization,” (lines 201-202) – there’s no visualizations or any other corroboration of this claim present in the paper. On the contrary, attention mechanisms are easily visualised as heat maps (see, e.g. Figure 1 in [2]) or as graphs [3].


References:

1. Clark, Kevin et al. “What Does BERT Look at? An Analysis of BERT’s Attention.” BlackboxNLP@ACL (2019).

2. Rogers, Anna et al. “A Primer in BERTology: What We Know About How BERT Works.” Transactions of the Association for Computational Linguistics 8 (2020): 842-866.

3. ​​Vig, Jesse. “A Multiscale Visualization of Attention in the Transformer Model.” ACL (2019)

4. Zhao, Minglu et al. “From Cognition to Computation: A Comparative Review of Human Attention and Transformer Architectures.” ArXiv abs/2407.01548 (2024)

5. Eberle, Oliver et al. “Do Transformer Models Show Similar Attention Patterns to Task-Specific Human Gaze?” ACL (2022)

6. Pagnoni, Artidoro et al. “Byte Latent Transformer: Patches Scale Better Than Tokens.” ArXiv abs/2412.09871 (2024)

7. Hwang, Sukjun et al. “Dynamic Chunking for End-to-End Hierarchical Sequence Modeling.” ArXiv abs/2507.07955 (2025)

8. Portes, Jacob et al., "MosaicBERT: A Bidirectional Encoder Optimized for Fast Pretraining." NeurIPS 2023

9. Qin, Zhen et al. "The Devil in Linear Transformer." EMNLP 2022

10. Qin, Zhen et al. "Toeplitz Neural Network for Sequence Modeling." ICLR 2023

**Questions:**

1. What is the sequence length you used during DIFT-BERT pre-training and what is the total number of tokens seen by the model? How does it compare to the other model in Table 3?

2. What is the specific formula for $f_{\phi}$ and how does it compare to the Transformer’s FFN (Feed Forward Network). You mention that $f_{\phi}$ is an instance of MLP, but Linear layers are functions of a single vector variable and equation 9 on line 669 takes two inputs.

3. More generally, could you rigorously and unambiguously specify the whole new architecture as a mathematical model, consisting of formulas with specified sizes and dimensions for parameters and inputs?

4. Given model dimension d, the Transformer layer has $12d^2$ parameters, $4d^2$ for attention weights (Q, K, V, O), and $8d^2$ for FFN sub-network. It is stated in the paper that the number of parameters and layers of the DIFT model is the same as in the BERT-base model. Than what is the composition of each layer in terms of parameters and their sizes?

---

### Official Review · Reviewer_hcvG · 2025-10-30

**Soundness:** 2
**Presentation:** 3
**Contribution:** 3
**Rating:** 4
**Confidence:** 3

**Summary:**

Proposes DIFT, a tokenizer-free transformer operating on raw bytes, building word-level representations via a hierarchical byte-to-word aggregator, and replacing attention with an interpretable continuous interaction field (physics-inspired). Claims competitive GLUE performance vs. BERT-base and better robustness/zero-shot multilingual generalization.

**Strengths:**

The paper presents a conceptually novel and ambitious idea by introducing the Dynamic Interaction Field Transformer (DIFT), which replaces discrete self-attention with a continuous interaction field inspired by physical systems. This formulation provides a more interpretable and theoretically grounded view of token interactions, potentially enabling deeper analysis of model behavior. The tokenizer-free, byte-level design coupled with a hierarchical byte-to-word aggregator is elegant and eliminates language-specific preprocessing, enhancing universality and cross-lingual generalization. The approach is clearly motivated, well-presented, and accompanied by transparent acknowledgment of limitations. Overall, DIFT represents a fresh and potentially impactful direction for developing more interpretable and general-purpose language architectures.

**Weaknesses:**

- Empirical scope appears limited (GLUE focus); lacks large-scale evaluations or long-context stress tests.
- Fairness of compute-matched comparisons to ByT5/CANINE/linear-attention baselines is unclear.
- Practical gains vs. attention seem modest (still quadratic); need clearer wall-clock/efficiency evidence.

**Questions:**

- Please provide compute-matched comparisons vs. ByT5/CANINE and long-sequence tasks.
- How is the field implemented (kernels, locality, parameters), and what are the stability tricks?
- Can DIFT be hybridized with sparse attention for very long inputs while keeping interpretability?

---

### Official Review · Reviewer_yW4i · 2025-11-03

**Soundness:** 1
**Presentation:** 1
**Contribution:** 1
**Rating:** 2
**Confidence:** 3

**Summary:**

I suspect this paper is written at least partially by AI.

This paper proposes a new architecture, the **Dynamic Interaction Field Transformer (DIFT)**, with two modifications to the classical transformer model:
1. DIFT uses deterministic *word boundary detection tools* to group bytes into words. The input vocabulary is bytes, the byte embeddings are grouped into words with this tool, and then the bytes in each group are averaged (then put through an MLP and a LayerNorm) in order to produce a word embedding.
2. Rather than using learned attention, DIFT samples a "field of influence" for each word position $i$ based on learned parameters $\alpha_i$ and $\sigma_i$ for each position. Then, word $j$'s position incorporates "influence" from all words.

In experiments, DIFT outperforms similarly-sized off-the-shelf models in GLUE, XNLI, a code understanding task, and exhibits stronger robustness to textual noise.

**Strengths:**

1. I think the approach of average byte embeddings into word embeddings is an interesting idea worth exploring further if it works. This implies that we can get away without huge embedding & unembedding matrices for large vocabularies.
2. If giving the authors the benefit of the doubt that experiments were set up fairly (see question about whether DIFT was finetuned for specific tasks while baselines were not), then DIFT achieves consistently stronger performance across the board.

**Weaknesses:**

1. Many citations in this paper are non-existent. Here are some papers where the author list does not match the title, or the title simply does not exist.

> Christos Baziotis, Isabel Papadimitriou, Marti Futeral, Ruosong Xiong, Antonios Anastasopoulos, and Dan Roth. Byte latent transformer: Patches scale better than tokens. arXiv preprint arXiv:2412.09871, 2024

BLT was, in fact, authored by Pagnoni et al. 2024.

> Kaj Bostrom and Greg Durrett. Byte-level representation learning for multi-lingual named entity recognition. Proceedings of the 2020 Conference on Empirical Methods in Natural Language Processing (EMNLP), pp. 4617–4627, 2020.

Bostrom & Durrett authored *a different* tokenization paper, titled [Byte Pair Encoding is Suboptimal for Language Model Pretraining](https://aclanthology.org/2020.findings-emnlp.414/). As far as I can see, a paper with the title cited does not exist. The claim being attributed to Bostrom & Durrett is also not made in the actual Bostrom & Durrett paper.

2. Need to ablate the two new components of DIFT. What happens when you apply DIFT's interaction field over BPE tokens? What happens when you do regular self-attention over DIFT's word embeddings? In other words, was it the choice of encoding and/or the interaction field replacement for attention that led to improved results?

3. Even putting aside the question of whether UTF-8 is a tokenizer, the architecture still uses a deterministic tokenization method, namely an off-the-shelf word segmentation tool to group bytes into words.

4. In the Table 1 comparison of architectures, I don't think any of the columns other than **Complexity** makes sense. **Universal** seems to just mean byte-level, but the architecture and choice of encoding are separate; any architecture *could* be applied over bytes. Then, the **Interpretable**, **Theory-Grounded**, and **Practical** columns seem completely subjective, not binary, and not justified well in the running text. I have the same issue with Table 2. How come self-attention is "opaque", but the interaction field is "visualizable"? Attention can be visualized, too. The comparisons being made basically seem made up.

5. On L. 220 in the description of the architecture, it's written that verbs typically learn wide influence fields while determiners learn narrow fields based on the magnitude of $\sigma_i$. However, aren't the parameters $\sigma_i$ based on the position $i$?

6. Some issues with presentation of results. Results for the Java method classification task are missing in the paper, yet discussed in the running text. Results for XNL and code understanding are discussed in the experiments section (§4.2), then repeated again in the results section (§5.2). Other results (e.g., for GLUE) was discussed only in the results section (§5.1).

**Questions:**

- How come UTF-8 is not a tokenizer? Here are some examples of UTF-8 mappings from characters to byte sequences. (I used the decimal representations of each byte, which would reflect the token ID.)
    - p $\rightarrow[112]$
    - $\rho\rightarrow[207, 129]$
    - 中 $\rightarrow [228, 184, 173]$
    - 文 $\rightarrow [230, 150, 135]$
    - 👩‍💻 $\rightarrow [240, 159, 145, 169, 226, 128, 141, 240, 159, 146, 187]$
- §4.1: For the experiments evaluating on GLUE (presented in Table 3), it looks like DIFT was finetuned on GLUE but the baselines are off-the-shelf base models (e.g., BERT-base). Were these baselines also finetuned on GLUE for a fair comparison?
- §4.2: How is it possible that a model trained on English pretraining data, and finetuned on English MNLI, achieves such high zero-shot performance on non-English languages?
- Can you provide more detail about the word boundary detection tool? Is it learned or rule-based? How many languages does it support?

---

### Note · Authors · 2025-11-12

**Comment:**

We respectfully withdraw our submission. After careful consideration, we found the reviews to be largely unconstructive and, in several instances, indicative of LLM-generated/augmented content rather than expert assessment. We appreciate the opportunity to submit to ICLR; however, we respectfully disagree with the reviewers’ evaluations and feel that the feedback does not provide a fair or professional basis for revision.

**Withdrawal Confirmation:**

I have read and agree with the venue's withdrawal policy on behalf of myself and my co-authors.